# A drug repurposing screen for whipworms informed by comparative genomics

Avril Coghlan[1‡], Frederick A. Partridge[2,3¤a‡], María Adelaida Duque-Correa[1¤b‡], Gabriel Rinaldi[1¤c‡], Simon Clare[1¤d], Lisa Seymour[1¤b], Cordelia Brandt[1], Tapoka T. Mkandawire[1], Catherine McCarthy[1], Nancy Holroyd[1], Marina Nick[2], Anwen E. Brown[2], Sirapat Tonitiwong[2], David B. Sattelle[2‡], Matthew Berriman[1¤e‡*]

1 Wellcome Sanger Institute, Cambridge, United Kingdom, 2 University College London, London, United Kingdom, 3 School of Life Sciences, University of Westminster, London, United Kingdom

¤a Current address: School of Life Sciences, University of Westminster, London, United Kingdom
¤b Current address: Wellcome-MRC Cambridge Stem Cell Institute, University of Cambridge, Cambridge, United Kingdom
¤c Current address: Department of Life Sciences, Aberystwyth University, Aberystwyth, United Kingdom
¤d Current address: Department of Medicine, University of Cambridge, Cambridge, United Kingdom
¤e Current address: School of Infection and Immunity, University of Glasgow, Glasgow, United Kingdom
‡ AC, FAP, MAD-C and GR share first authorship on this work. DBS and MB are joint senior authors on this work.
* Matt.Berriman@glasgow.ac.uk

**Data Availability Statement:** All relevant data are within the paper and its Supporting Information files.

## Abstract

Hundreds of millions of people worldwide are infected with the whipworm *Trichuris trichiura*. Novel treatments are urgently needed as current drugs, such as albendazole, have relatively low efficacy. We have investigated whether drugs approved for other human diseases could be repurposed as novel anti-whipworm drugs. In a previous comparative genomics analysis, we identified 409 drugs approved for human use that we predicted to target parasitic worm proteins. Here we tested these *ex vivo* by assessing motility of adult worms of *Trichuris muris*, the murine whipworm, an established model for human whipworm research. We identified 14 compounds with $EC_{50}$ values of $\leq$50 µM against *T. muris ex vivo*, and selected nine for testing *in vivo*. However, the best worm burden reduction seen in mice was just 19%. The high number of *ex vivo* hits against *T. muris* shows that we were successful at predicting parasite proteins that could be targeted by approved drugs. In contrast, the low efficacy of these compounds in mice suggest challenges due to their chemical properties (e.g. lipophilicity, polarity, molecular weight) and pharmacokinetics (e.g. absorption, distribution, metabolism, and excretion) that may (i) promote absorption by the host gastrointestinal tract, thereby reducing availability to the worms embedded in the large intestine, and/or (ii) restrict drug uptake by the worms. This indicates that identifying structural analogues that have reduced absorption by the host, and increased uptake by worms, may be necessary for successful drug development against whipworms.

**Funding:** The research in MB's group at the Wellcome Sanger Institute was funded by the Wellcome Trust (https://wellcome.org/) [grant number 206194 to the Wellcome Sanger Institute]. DBS and FAP were supported by a Medical Research Council (https://www.ukri.org/councils/mrc/) grant MR/N024842/1, and a UCL (https://www.ucl.ac.uk/) Therapeutic Innovation Networks Pilot Data Scheme award supported by funding from MRC UCL Confidence in Concept (CiC6) 2017 MC_PC_17180. The funders had no role in study design, data collection and analysis, decision to publish, or preparation of the manuscript.

**Competing interests:** The authors have declared that no competing interests exist.

## Author summary

Our study describes a drug repurposing screen against the whipworm *Trichuris*, which causes the neglected tropical disease trichuriasis. Despite the pressing need for more effective drugs to treat whipworm infections, *Trichuris* has been the focus of extremely few drug screens. Using a combination of comparative genomics, and data on drugs and their targets from the ChEMBL database, we predicted that 409 drugs approved for human use would target *Trichuris* proteins. Using a high-throughput screening platform, we then screened these 409 drugs against *Trichuris adults ex vivo*. Our high hit rate of 12% demonstrated the utility of our comparative genomics approach to predict worm proteins for targeting by approved drugs. We subsequently tested the most active compounds in *Trichuris*-infected mice, but there were no significant hits *in vivo*. Interestingly, this accorded with a previously published screen against *Trichuris*, which also reported promising *ex vivo* hits but low activity in mice. We discuss possible reasons for this finding, and important implications for optimising future drug screens against *Trichuris*.

## Introduction

An estimated 316–413 million people worldwide are infected with whipworms (*Trichuris trichiura*), the cause of the neglected tropical disease trichuriasis [1]. Whipworms infect the large intestine, causing abdominal pain, tiredness, colitis, anaemia and *Trichuris* dysentery syndrome [2]. In addition, chronic infection of children with whipworms is associated with impaired cognitive and physical development [2]. There is an urgent need for new treatments to fight whipworm infection because no vaccines are available and single doses of mebendazole or albendazole, the mainstay of mass drug administration programmes, do not achieve complete deworming [3]. Indeed, of the three major soil-transmitted helminthiases, current drugs have the lowest efficacy for trichuriasis [4,5]. Furthermore, there is worrying evidence of resistance to mebendazole and albendazole in *Trichuris* in humans [6,7].

Drug discovery for trichuriasis and other parasitic nematodes is challenging given the absence of *ex vivo* culture systems. Caecal organoids are a promising platform for *in vitro*/*ex vivo* culture of whipworms [8], but are still in the early stages of development. On the other hand, parasite maintenance in mice is expensive, labour-intensive, and has ethical costs. In addition, since there is also a great need for new treatments for other soil-transmitted nematodes (e.g. *Ascaris*, hookworms), ideally one wants compounds effective against several parasitic species. Furthermore, even if promising hits are found in a screen, because soil-transmitted nematodes affect low income countries, there are financial challenges for pharmaceutical companies to bring an effective molecule to market, which requires years of investment to characterise its pharmacology and safety [5].

Drug repurposing aims to test currently approved drugs for new uses such as trichuriasis, so that any promising hits can progress relatively quickly through the drug development pipeline, since a lot is already known about their safety and pharmacology [9]. One drug repurposing approach, taken by the ReFRAME project, is to test all approved drugs as well as compounds that have undergone significant pre-clinical studies, to find potential new drugs against pathogens [10].

We have previously taken an alternative approach to identify potential drugs for repurposing [11]. Comparative genomics of many different parasitic worms, including *Trichuris* spp., was used to identify proteins that are homologous to known drug targets from other species and then to select known ligands of those targets as candidates with possible activity against

worms. Making use of the ChEMBL database of bioactive molecules and their targets [12], we identified drugs approved for human use that we predicted to interact with protein targets in parasitic worms. Briefly, we prioritised worm proteins that had high similarity to their top BLASTP match in ChEMBL, that lacked human homologues, or had *Caenorhabditis elegans/ Drosophila melanogaster* homologues with lethal phenotypes [11]. For each worm protein, we retrieved compounds with high potency/affinity for its top ChEMBL target. After collapsing compounds within a single chemical class to one representative, the resultant set included 817 approved drugs, of which we purchased as many as possible within our budget (prioritising the cheapest), i.e. 409 compounds.

Here, we screened these 409 drugs against adult worms of *T. muris*, the natural mouse whipworm and model of the closely related human parasite *T. trichiura*. Our screen was based upon quantifying worm motility *ex vivo* in the presence of the drugs, using the Invertebrate Automated Phenotyping Platform (INVAPP) [13,14]. This is an attractive approach since several drugs that have been used in the past to treat trichuriasis, such as levamisole, pyrantel, and oxantel [15], cause spastic paralysis of the worms, thereby facilitating their expulsion (while still alive) by peristalsis of the host intestine [16]. For the highest scoring hits, we then performed *ex vivo* dose-response experiments to estimate $EC_{50}$ (half maximal effective concentration). The most active compounds were subsequently tested in *T. muris*-infected mice. Interestingly, there were a high number of hits *ex vivo*, but no significant hits *in vivo*; we discuss possible reasons for this finding, and its implications for future drug screens against whipworms.

## Methods

### Ethics statement

Mouse experimental infections were conducted under the UK Home Office Project Licence No. PPL (P77E8A062). All protocols were revised and approved by the Animal Welfare and Ethical Review Body (AWERB) of the Wellcome Sanger Institute. The AWERB is constituted as required by the UK Animals (Scientific Procedures) Act 1986 Amendment Regulations 2012.

### Drugs

Based on a comparative genomics analysis, we previously proposed a screening set for parasitic worms, of 817 drugs approved for human use [11]. Taking into account price, availability and chemical class, 409 compounds were obtained from Sigma-Aldrich in 10 mM dimethylsulfoxide (DMSO) solutions, and stored at -20˚C (**S1 Table** and **Fig 1**). For *in vivo* testing, we obtained high-purity compounds from Sigma-Aldrich: econazole nitrate (catalogue number E0050000), prazosin hydrochloride (BP399), flunarizine dihydrochloride (F0189900), butoconazole nitrate (1082300), felodipine (BP777), cyproheptadine hydrochloride (1161000), terfenadine (T00710000), pimozide (BP682 or P1793), nicardipine hydrochloride (1463224 or N7510), mebendazole (1375502), as well as Tween 80 (P6474) and ethanol (PHR1373) for dissolving the drugs.

### Mice

NOD SCID mice (NOD.*Cg-Prkdc*scid Il2rgtm1Wjl/SzJ) were used to maintain the life cycle of *T. muris* and for *in vivo* testing of drugs. All animals were housed in GM500 Individually Ventilated Cages or IsoCage N-Biocontainment Systems (Tecniplast) under environmentally-controlled conditions (temperature: 19–23˚C, humidity: 45–65%, light/dark cycle 12h/12h), with

Obtained 409 approved phase III/IV drugs

↓

Tested *in vitro* at 100 µM for 24 h

↓

Re-tested top 50 compounds at 100 µM, 20 µM, and 5 µM

↓

Identified & obtained 71 structurally similar inexpensive or over-the-counter drugs, and tested them *in vitro*

↓

Identified 11 'top priority compounds', and 10 'second priority' compounds & generated *in vitro* dose response curves for them

↓

Identified 10 candidates for *in vivo* testing in mice: drugs with $EC_{50} < 50$ µM *in vitro*

↓

Decided dosage in mice for each drug (15-100 mg/kg), based on published LD50, & previous dosages with no adverse effects in published mice studies

↓

Tested 9 drugs for anthelmintic efficacy *in vivo* in mice

**Fig 1. Study flow for testing 409 approved drugs against *T. muris*.**

access to water and rodent food. No more than five animals were housed per cage. Welfare assessments were carried out daily, and abnormal signs of behaviour or clinical signs of concern were reported.

## Parasites

Mouse infections to maintain *T. muris* were conducted as previously described [17]. Female NOD SCID mice (6 wk old) were infected with a dose of 400 infective embryonated *T. muris* eggs via oral gavage. Thirty-five days later, the mice were euthanised and their caecae and proximal colons removed. Intestines were opened longitudinally, washed with pre-warmed Roswell Park Memorial Institute (RPMI)-1640 media supplemented with penicillin (500 U/mL) and streptomycin (500 µg/mL) (all from Sigma-Aldrich), and adult worms were carefully removed using forceps. Worms were maintained in RPMI-1640 media supplemented with penicillin (500 U/mL) and streptomycin (500 µg/mL) at approximately 37˚C. On the same day of collection, worms were sent by courier in flasks at 37˚C to UCL.

Ex vivo T. muris *drug screen*. At UCL, individual adult worms were placed in wells of 96-well plates containing 100 µL of RPMI-1640 media supplemented with penicillin (500 U/mL) and streptomycin (500 µg/mL), plus the tested compound dissolved in 1% v/v DMSO final concentration. Levamisole (10 µM, Sigma-Aldrich) was used as a positive control, and 1% DMSO as a negative control.

Plates were incubated at 37˚C and 5% $CO_2$. Motility was determined after 24 h using the Invertebrate Automated Phenotyping Platform (INVAPP) [13,14], which recorded 200-frame movies of the whole plate at 100 ms intervals. The Paragon algorithm [13] was used to detect changes in motility by analysing changes in pixel variance.

In the primary library screen, the test drug concentration was 100 µM, and each compound was screened in triplicate using three adult worms in three individual wells, carried out over

three separate occasions. We previously found that 100µM is a good concentration for an initial screen against *Trichuris* using our INVAPP screening platform [18]. In the confirmatory re-screen, each of the top 50 hits from the primary screen was tested again at 100 µM in six replicates (using six adult worms in six individual wells) (**Fig 1**). Each replicate used worms obtained from independent mice, and the re-screen was carried out over two occasions. To help prioritise the hit compounds for further investigation, their activity was evaluated at 5 and 20 µM, with six replicates for each compound.

Caenorhabditis elegans *drug screen.* To obtain *C. elegans* strain N2 worms for screening, synchronised L1 worms were prepared as described in [13]. *C. elegans* var Bristol, originally isolated from mushroom compost near Bristol in the UK, is referred to as N2. In the screen, worms were cultured at 20˚C for 6 days in 96-well plates containing *E. coli* food, with approximately 20 L1s per well, and 100 µM of tested drug (1% v/v DMSO) or a diluted DMSO control, as above. The worms were then imaged using INVAPP and a motility score calculated using Paragon, as for the *T. muris* screen. The top 14 hits from the primary library screen based on the lowest mean motility score were re-screened at 100 µM, with six replicates for each compound.

## Cheminformatics analysis of the top 50 hits

The top 50 hits from the primary library screen in *T. muris* (based on the lowest mean motility score) were assigned to chemical classes, based on information in DrugBank [19], ChEBI [20], Wikipedia, and PubMed (**S1 Fig**). We identified additional approved drugs in ChEMBL v25 [12] that were similar to our top 50 hits, using two approaches in DataWarrior v5.0.0 [21]: (i) the 'Similarity analysis' function, and (ii) the 'Substructure search' function, to search for compounds with substructures present in our top 50 hits (**S2 Fig**). The additional drugs identified were filtered to only retain over-the-counter or inexpensive drugs. Drugs were considered inexpensive if they cost $\leq$ US $5 per vial/capsule according to DrugBank v5.1.4 [19], while over-the-counter drugs (which are usually very safe) were identified from ChEMBL [12]. We discarded antipsychotics, general anaesthetics and anti-clotting agents, due to possible low suitability for repurposing. This led us to obtain an additional 71 drugs from Sigma-Aldrich for testing (**Fig 1**).

Ex vivo T. muris *drug screen on additional approved drugs.* The 71 additional approved drugs were first prioritised by measuring their activity in an indicative screen with four adult worms in four individual wells: one at 100 µM, two at 50 µM, and one at 20 µM. This did not allow us to determine activity/non-activity of these compounds but rather enabled us to focus our investigation on compounds of interest. Seventeen compounds were selected and were tested at 100 µM and 50 µM for 24 h, with six replicates each.

$EC_{50}$ *values* ex vivo *in* T. muris. To determine the relative potency of the most promising drugs, activity was measured at eight concentrations (typically between 100 µM and 10 nM). The motility of five individual worms obtained from different mice was measured at each point. Using the R package drc [22], concentration-response curves were fitted using the three-parameter log-logistic model, and $EC_{50}$ values estimated for the drugs.

In vivo *drug screen against* T. muris. Nine drugs were tested in mice, following a similar protocol to [23]. Briefly, female NOD SCID mice (6 wk old) were infected with a low dose (approximately 20–30 eggs) of *T. muris* eggs via oral gavage, with six mice per treatment group. To confirm establishment of infection, on day 35 post-infection (D35 p.i.), faecal pellets from individual mice were collected and faecal smears performed to check for the presence of *T. muris* eggs. Briefly, each mouse was temporarily placed in a beaker and a faecal sample collected using forceps. Half of the faecal pellet was placed on a glass slide, ~50 µL of UltraPure

water added and the pellet gently squashed with a second slide. The sample was then observed under the microscope (X40 objective) and classified as egg-positive or egg-negative.

Treatment with mebendazole (positive control), vehicle-only (negative control), or candidate drugs, was performed by oral gavage, for three consecutive days on D36, D37 and D38 p. i.. We used a three-day treatment regimen (following [24]) because mebendazole treatment for three days is far more effective in humans than a single dose [25].

The compounds were administered at dosages of 15–100 mg/kg of body weight, which were equivalent to ~2–16% of their $LD_{50}$, and were previously reported to cause no adverse effects on mice (**S2 Table**). A dose of 50 mg/kg body weight was used for mebendazole. The drug vehicle consisted of 7% Tween 80, 3% ethanol and 90% water (v/v/v), following previous mouse drug screens [26,27]. At D44 p.i. the mice were culled and intestines dissected for adult worm collection and counting. The worm burden (WB) of treated mice was compared with the WB of control (vehicle-only) mice. The worm burden reduction (WBR) was calculated as: WBR (%) = 100%—(100% * WB-treatment/WB-control).

## Statistics

For the confirmatory *ex vivo* re-screens in *T. muris* and *C. elegans*, a P-value was calculated for each compound using a Mann-Whitney test to compare motility scores in drug-treated wells to the DMSO control wells. These P-values were corrected for multiple testing using the Bonferroni correction. For the *in vivo* drug testing, the Wilcoxon test (in R) was used to determine the statistical significance of WBRs.

## Results

*Worm motility* ex vivo *was greatly reduced by the top 50 hit compounds*. We screened 409 drugs approved for human use by evaluating their effect on motility of *T. muris* adult worms *ex vivo*, using the automated system INVAPP [13] and a drug concentration of 100μM for 24 h (**Fig 1**). Some of the drugs tested (e.g. antipsychotics) may be unsuitable for repurposing, but were included in the hope that if they were hits, we would next identify structurally similar, but safer, approved drugs (e.g. antihistamines) to test. The distribution of effects of the drugs on motility are shown in **Fig 2A**, and the motility score for each drug given in **S1 Table**. Complete loss of motility was observed for 26 drugs, while another 24 compounds induced a partial reduction in motility.

The chemical and pharmacological features of these top 50 drugs are summarised in **Table 1** (with additional details in **S3 Table**). They included compounds in 24 broad chemical classes (**S1 Fig**).

To confirm activity, the top 50 hits were re-screened at 100 μM for 24 h. Of the 50, 45 showed a significant reduction in parasite motility compared with the negative (DMSO-only) control ($P<0.05$) (**Fig 2B** and **S3 Table**). The effects on motility for the DMSO-only control and two exemplar hits that block movement, astemizole and pimozide, are shown in the form of time-lapse frames in **Fig 2D**, and as a full recording in **S1 Movie**. To help prioritise the hit compounds for further investigation, their relative potency was assessed by also estimating their activity at lower concentrations (5 and 20 μM for 24 h). Only pimozide, nicardipine, terfenadine, thonzonium, nicotine, paraoxon and dicumarol were clearly associated with reduced motility ($P<0.05$) at these lower concentrations (**S3 Table**), although the latter three were excluded from subsequent screens due to unfavourable properties (paraoxon, incorrectly classified as an approved drug; nicotine, neurotoxic; dicumarol, blood clotting effects).

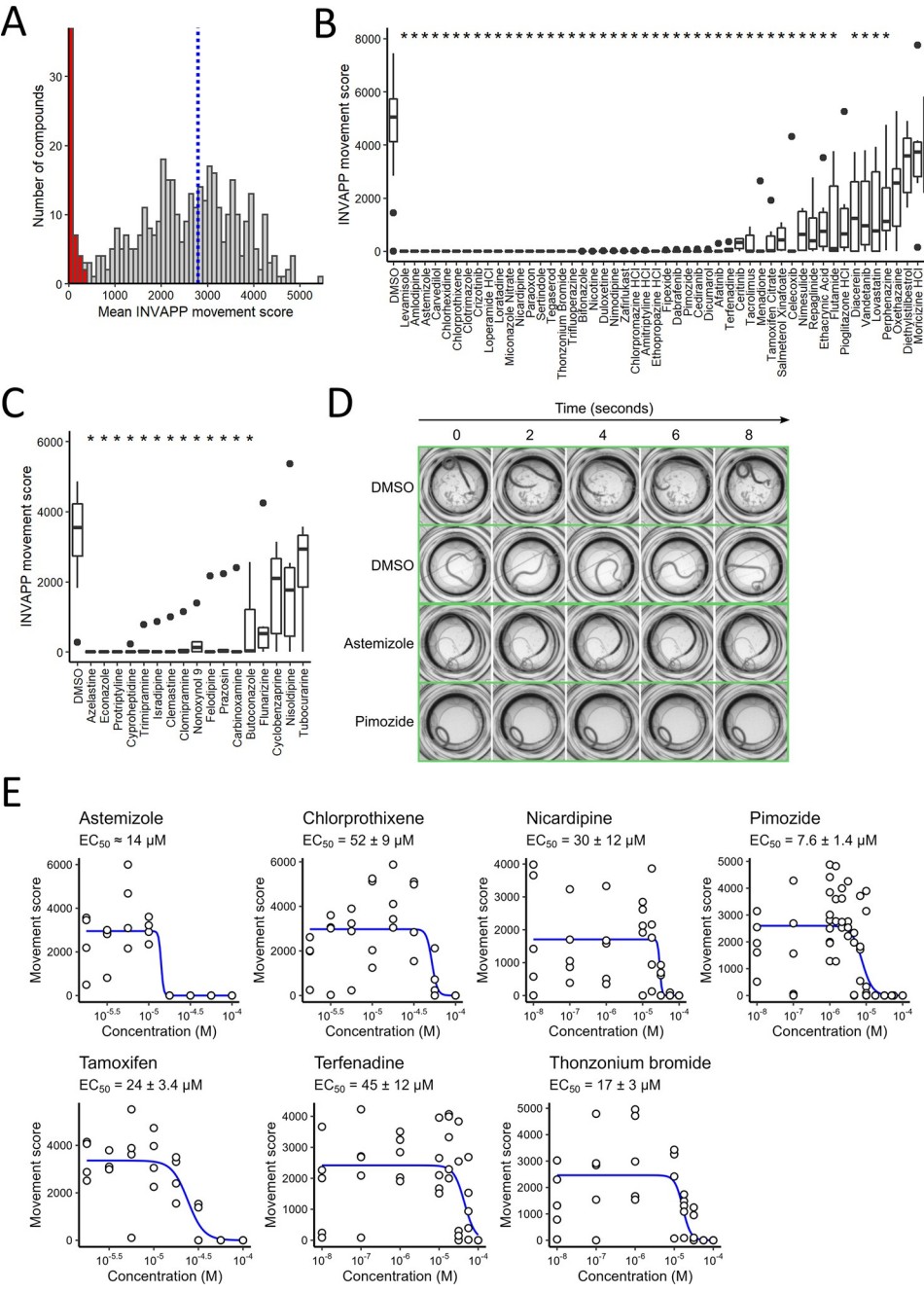

**Fig 2. Ex vivo T. muris *drug screen*.** (A) Histogram showing the mean effect on *T. muris* adult motility of each of the 409 approved drugs tested in the primary screen at 100 µM. Blue dotted line indicates the mean movement score for DMSO-only negative controls. The 50 compounds with the greatest reduction in mean movement score, indicated in red, were selected as candidate hits. (B) Secondary screen confirmed the activity of 46 drugs at 100 µM. * indicates significant reduction in *T. muris* adult movement score compared to the DMSO-only control (Mann-Whitney test adjusted for multiple comparisons by the Bonferroni method, $P < 0.05$, $n = 6$). (C) Identification of an additional 13 active drugs by testing structurally-related drugs in the same *T. muris* adult motility assay at 100 µM. * indicates significant reduction in *T. muris* adult movement score compared to the DMSO-only control (Mann-Whitney test adjusted for multiple comparisons by the Bonferroni method, $P < 0.05$, $n = 6$). (D) Montage of frames selected from time-lapse recording of worms treated with DMSO alone, or with astemizole or pimozide. A movie of this data is presented in **S1 Movie**. (E) Concentration-response curves for selected active drugs with $EC_{50}$ values at or below 50 µM. Curves fitted with the three-parameter log-logistic model. $EC_{50}$ values are indicated ± the standard deviation. For astemizole, there was large uncertainty in the calculated $EC_{50}$ value, so it is shown as an estimate.

**Table 1. The top 50 hits in our initial *ex vivo* screen of approved drugs (at 100 μM for 24 hr) against *T. muris* adults, and the top 17 hits in our screen of structurally related drugs.**

| A. Chemical class | B. Name of drug | C. Use in humans | D. Target for use in humans | E. Nerve/ muscle | F. Properties | G. *Ex vivo* anthelmintic activity (*in vivo* for *C. elegans*) |
|---|---|---|---|---|---|---|
| Dihydropyridines | nicardipine | hypertension | blocks $Ca^{2+}$ channels | yes | **In** | **Sm** [27,28] |
| Dihydropyridines | nitrendipine | hypertension | blocks $Ca^{2+}$ channels | yes | | |
| Dihydropyridines | nimodipine | hypertension | blocks $Ca^{2+}$ channels | yes | **B** | **Acy** [29] |
| Dihydropyridines | amlodipine | hypertension | blocks $Ca^{2+}$ channels | yes | **B, In, WHO** | **Sm** [27] |
| Serotonin-like | fipexide | senile dementia | may increase neurotransmitter activity | yes | **W** | |
| Serotonin-like | astemizole | allergic disorders | histamine receptor antagonist | yes | **W** | |
| Serotonin-like | tegaserod | irritable bowel syndrome | serotonin receptor agonist/ antagonist | yes | **W** | |
| Serotonin-like | sertindole | psychotic disorders | dopamine/serotonin receptor antagonist | yes | **W** | |
| Serotonin-like; Piperidines | pimozide | psychotic disorders | dopamine/serotonin receptor antagonist | yes | **In** | **Cel** [30], **Sm** [27,31] |
| Piperidines | terfenadine | allergic disorders | histamine receptor antagonist | yes | **W** | **Sm** [27] |
| Piperidines | loperamide | diarrhoea | blocks $Ca^{2+}$ channels; opioid receptor agonist | yes | **B, In, WHO, OC** | **Egr** [32] |
| Surfactants | thonzonium | in skin/nasal drops | disperses cellular debris* | | **Top** | **Acy** [23], **Sm** [27] |
| Alpha/beta receptor ligands | duloxetine | depression; anxiety | inhibits noradrenaline/serotonin re-uptake | yes | **B, In** | |
| Alpha/beta receptor ligands | carvedilol | cardiovascular disorders | alpha/beta receptor antagonist | yes | **In, WHO** | |
| Alpha/beta receptor ligands | salmeterol | asthma | beta receptor agonist | yes | **Top, B, In** | |
| Imidazoles | clotrimazole | fungal infections | inhibits fungal ergosterol production* | | **In, WHO, OC** | **Cel** [30], **Ofel** [33] |
| Imidazoles | bifonazole | fungal infections | inhibits fungal ergosterol production* | | **Top** | **Acy** [23] |
| Imidazoles | miconazole | fungal infections | inhibits fungal ergosterol production* | | **B, In, WHO, OC** | **Cel** [34], **Sm** [35,36], **Ofel** [33] |
| Phenothiazines | perphenazine | psychotic disorders | dopamine receptor antagonist | yes | **B, In** | **Sm** [37] |
| Phenothiazines | chlorpromazine | psychotic disorders | dopamine/serotonin/histamine receptor antagonist | yes | **B, In, WHO** | **Cel** [30], **Acan, Tmu, Sm** [34,35,37], **Acy** [23] |
| Phenothiazines | chlorprothixene | psychotic disorders | dopamine/serotonin/histamine receptor antagonist | yes | | **Cel** [30], **Acy** [23], **Sm** [27,37] |
| Phenothiazines | trifluoperazine | psychotic disorders | dopamine/serotonin receptor antagonist | yes | **B, In** | **Cel** [30], **Sm** [37,38] |
| Phenothiazines | ethopropazine | Parkinson's disease | histamine and muscarinic receptor antagonist | yes | **In** | **Acy, Hpol, Tmu** [23] |
| Phenothiazines | moricizine | irregular heartbeats | blocks $Na^+$ channels | yes | **B** | |
| Oestrogen receptor ligands | diethylstilbestrol | prostate cancer | oestrogen receptor agonist | | **W** | |
| Oestrogen receptor ligands | tamoxifen | breast cancer | inhibits oestrogen binding to its receptor | | **B, In, WHO** | **Sm** [27] |
| Kinase inhibitors | crizotinib | non-small cell lung cancer | inhibitor of receptor tyrosine kinases | | | **Sm** [26] |
| Kinase inhibitors | afatinib | non-small cell lung cancer | inhibitor of tyrosine kinases | | **WHO** | **Sm** [26] |
| Kinase inhibitors | vandetanib | thyroid cancer | inhibitor of tyrosine kinases | | | **Sm** [26] |
| Kinase inhibitors | cediranib | cancers e.g. liver cancer | inhibitor of receptor tyrosine kinases | | | |

*(Continued)*

**Table 1.** (*Continued*)

| A. Chemical class | B. Name of drug | C. Use in humans | D. Target for use in humans | E. Nerve/ muscle | F. Properties | *G. Ex vivo* anthelmintic activity (*in vivo* for *C. elegans*) |
|---|---|---|---|---|---|---|
| Kinase inhibitors | ceritinib | non-small cell lung cancer | inhibitor of receptor tyrosine kinases | | | |
| Kinase inhibitors | dabrafenib | thyroid cancer; melanoma | inhibitor of kinases | | | |
| Biguanides | chlorhexidine | skin antiseptic agent | disrupts microbial cell membranes* | | **Top, In, WHO, OC** | **Acy** [23] |
| Local anaesthetics | oxethazaine | pain from stomach ulcers | local anaesthetic effect on the gastric mucosa | yes | | **Sm** [27] |
| Meglitinides | repaglinide | type 2 diabetes mellitus | closes $K^+$ channels in pancreatic beta-cells | | **In** | |
| Dibenzocycloheptenes | amitriptyline | depression | inhibits noradrenaline/serotonin re-uptake | yes | **B, In, WHO** | **Acy** [23], **Sm** [35,37] |
| Dibenzocycloheptenes | loratadine | allergic disorders | histamine receptor antagonist | yes | **In, WHO, OC** | |
| Loop diuretics | ethacrynic acid | hypertension | inhibits the $Na^+ K^+ Cl^-$ cotransporter 2 | | **In** | **Sdig** [39] |
| Benzenesulfonamides | zafirlukast | asthma | leukotriene receptor antagonist | yes | **In** | |
| Vitamin K-like | menadione | nutritional supplement | precursor of vitamin K | | | **Sm** [27] |
| Vitamin K-like | dicumarol | decreasing blood clotting | inhibits vitamin K reductase | | | |
| Anthraquinones | diacerein | osteoarthritis | reduces interleukin-1 beta activity | | | |
| Macrolides | tacrolimus | organ transplants | inhibits immunophilin FKBP | | **B, In** | |
| COX-2 inhibitors | celecoxib | arthritis pain | inhibits COX-2 | | **W** | **Sm** [26] |
| COX-2 inhibitors | nimesulide | acute pain; osteoarthritis | inhibits COX-2 | | **W** | |
| Antiandrogens | flutamide | carcinoma of the prostate | androgen receptor antagonist | | **B, In** | **Cel** [30] |
| Thiazolidinediones | pioglitazone | type 2 diabetes | nuclear receptor PPAR-gamma agonist | | **B** | |
| Statins | lovastatin | to lower risk of heart attack | inhibits HMG-CoA reductase | | **In** | **Sm** [40] |
| Toxic alkaloids | nicotine | nicotine withdrawal | nicotinic cholinergic receptor agonist | yes | **Tox** | **Acy** [29]; did not search literature for more |
| Insecticides | paraoxon | not a drug; insecticide | inhibits acetylcholinesterases | yes | **Tox** | did not search literature |
| Piperidine-like | flunarizine | migraine; vascular disease | calcium channel blocker | yes | **In** | **Sm** [27] |
| Piperidine-like | carbinoxamine | allergic disorders | histamine receptor antagonist | yes | **In** | |
| Piperidine-like | clemastine | allergic disorders | histamine receptor antagonist | yes | **OC, In** | **Acy, Hpol, Tmu** [23] |
| Dibenzocycloheptenes | clomipramine | depression; OCD | inhibitor of serotonin re-uptake | yes | **In, WHO, B** | |
| Dibenzocycloheptenes | cyclobenzaprine | muscle spasm | serotonin receptor antagonist | yes | **In** | |
| Dibenzocycloheptenes | cyproheptadine | allergic disorders | serotonin/histamine receptor antagonist | yes | **In** | **Acy** [23] |
| Dibenzocycloheptenes | trimipramine | depression | serotonin/histamine receptor antagonist | yes | **In, B** | **Acy, Hpol, Tmu** [23] |
| Dibenzocycloheptenes | protriptyline | depression | inhibitor of serotonin/ noradrenaline re-uptake | yes | **In, B** | **Acy, Tmu** [29] |
| Imidazoles | butoconazole | fungal infections | inhibits fungal ergosterol production* | | **OC, In, Top** | |
| Imidazoles | econazole | fungal infections | inhibits fungal ergosterol production* | | **In, Top** | **Cel** [30]; **Acy, Tmu** [29] |

(*Continued*)

**Table 1.** (Continued)

| A. Chemical class | B. Name of drug | C. Use in humans | D. Target for use in humans | E. Nerve/ muscle | F. Properties | *G. Ex vivo* anthelmintic activity (*in vivo* for *C. elegans*) |
|---|---|---|---|---|---|---|
| Dihydropyridines | isradipine | hypertension | blocks $Ca^{2+}$ channels | yes | **In** | **Cel** [13] |
| Dihydropyridines | felodipine | hypertension | blocks $Ca^{2+}$ channels | yes | **In** | **Cel** [30] |
| Dihydropyridines | nisoldipine | hypertension | blocks $Ca^{2+}$ channels | yes | **In** | |
| Alpha/beta receptor antagonists | prazosin | hypertension | alpha receptor antagonist | yes | **In** | **Sm** [27] |
| Kinase inhibitor-like | tubocurarine | muscle relaxation | Ach receptor antagonist | yes | **In; by injection** | |
| Antihistamine | azelastine | rhinitis; conjunctivitis | histamine receptor antagonist | yes | **In; Top** | |
| Thonozonium-like | nonoxynol 9 | surfactant in spermicide | surfactant* | | **OC; Top** | |

The first 50 drugs are the top 50 hits in the initial screen, and the last 17 are the top 17 hits in our screen of structurally related drugs. Column E says whether the mode of action of the current use in human affects the nervous system or muscle. In column F, **Top** means only approved for topical use (based on data in ChEMBL and DrugBank [19]; **B** means a 'black box' warning; **W** means withdrawn; **Tox** denotes highly toxic compounds; **In** denotes inexpensive compounds (≤US \$5 per vial/ capsule, based on data in DrugBank); **WHO** denotes a compound on the WHO list of essential medicines (from https://www.who.int/medicines/publications/ essentialmedicines/en/); and **OC** a drug sold over-the-counter (data from ChEMBL). The salt form tested is given in **S1 Table**. In column G, species of helminths are abbreviated as: Sm = *Schistosoma mansoni*, Ofel = *Opisthorchis felineus*, Egr = *Echinococcus granulosus*, Acy = *Ancylostoma ceylanicum*, Acan = *Ancylostoma caninum*, Cel = *Caenorhabditis elegans*, Tmu = *Trichuris muris*, Hpol = *Heligmosomoides polygyrus*, Sdig = *Setaria digitata*.

\*The target of the imidazoles in nematodes may be cytochrome CYP-450 [29] and/or a nicotinic acetylcholine receptor [41]; while the surfactants thonzonium, nonoxynol and chlorhexidine may disrupt the cell membranes of nematode epidermal cells.

## Expansion screen using structurally related approved drugs for hit compounds

Having identified hit compounds from a variety of chemical classes, we wanted to ensure that we had found the most active compounds from each class to study further. We therefore searched ChEMBL [12] for additional approved drugs that are structurally related to our top 50 hits. This led us to obtain an additional 71 approved drugs (**S4 Table** and **Fig 1**). Amongst the top 50 hits were several antihistamines, which tend to be very safe, so we chose to further investigate two antihistamines (azelastine, cyproheptadine) that were just below our cut-off for the top 50 hits in the primary library screen.

We prioritised the 71 additional compounds and two antihistamines, by first estimating their relative activity against *T. muris ex vivo* in a small-scale pilot experiment at 100 μM, 50 μM and 20 μM (**S4 Table, columns O, P, Q**). Based on this, 17 compounds were selected to be re-tested at 100 μM and 50 μM (**S3 Fig**). Of the 17 drugs, 13 significantly reduced motility ($P<0.05$) at 100 μM; and 2 significantly reduced motility at 50 μM, with flunarizine and nonoxynol-9 just above the significance threshold ($P = 0.056$ and $P = 0.061$, respectively) (**S4 Table** and **Fig 2C**).

### *Only two hit compounds were effective against both* C. elegans *and* T. muris

*C. elegans* is a free-living nematode but commonly used as a model for parasitic nematodes [5,42]. The 409 approved drugs were screened at 100 μM in a *C. elegans* assay that followed worm development from L1 to L4/young adult stages to identify those compounds that either slowed/blocked larval development or reduced motility. Far fewer hits were detected than in the *T. muris* screen; the top 14 candidates were re-screened at 100 μM, and five significantly reduced the growth/motility of *C. elegans* ($P<0.05$): ponatinib, chlorhexidine, clotrimazole,

sorafenib, and paroxetine. Thus, of the 409 approved drugs screened, only clotrimazole and chlorhexidine were hits in both *T. muris* and *C. elegans* (**S4 Fig**). Clotrimazole and other imidazoles may have cytochrome CYP-450 as a conserved target across fungi, flatworms, and nematodes [29,36], or alternatively may target a nicotinic acetylcholine receptor in nematodes [41]. Chlorhexidine, a surfactant, may disrupt the cell membranes of nematode epidermal cells, as it does for bacterial cells. Three antihistamines were hits in *T. muris*, but there were no antihistamine hits in *C. elegans*.

### Ex vivo *potency guided the selection of drugs for* in vivo T. muris efficacy tests

Based on the *ex vivo* motility results, we selected the seven most active drugs (pimozide, astemizole, thonzonium, tamoxifen, nicardipine, terfenadine, and chlorprothixene) and determined their relative potency by measuring concentration-response curves across eight concentrations using the *T. muris ex vivo* motility assay. The $EC_{50}$ of six drugs was below 50 μM and that of the seventh, chlorprothixene, was $52 \pm 9$ μM (**Fig 2E** and **S5 Table**). For an additional seven drugs (econazole, cyproheptadine, butoconazole, flunarizine, clemastine, felodipine, and prazosin) we did not measure the $EC_{50}$ using a full concentration-response curve. However, based on single-concentration activity measurements at 5, 20, 50 and 100 μM in the drug screens described earlier (see previous paragraphs), we estimated that the $EC_{50}$ of these drugs was at or below 50 μM (**S5 Table**).

Of the 14 compounds with $EC_{50}$ of $\leq 50$ μM, we considered thonzonium unsuitable for testing in mice because it is only approved for topical use in humans; and tamoxifen and chlorprothixene unsuitable because of the potential for serious side effects. Clemastine was previously tested against *T. muris* in mice by [23], who found it resulted in a worm burden reduction of 20.1%. The remaining ten drugs (pimozide, astemizole, nicardipine, terfenadine, econazole, cyproheptadine, butoconazole, flunarizine, felodipine, and prazosin) were considered good candidates to test in mice. Although pimozide is an antipsychotic, it has also been used for non-psychotic disorders such as tics [43], so we tested it in mice in the hope that a low dose could reduce worm burden without serious side effects. Although econazole and butocondazole are only approved for topical use in humans, they have been previously tested orally in mice (e.g. [44]). We were unable to obtain astemizole, so tested nine drugs in mice (**S5 Table** and **Fig 3**).

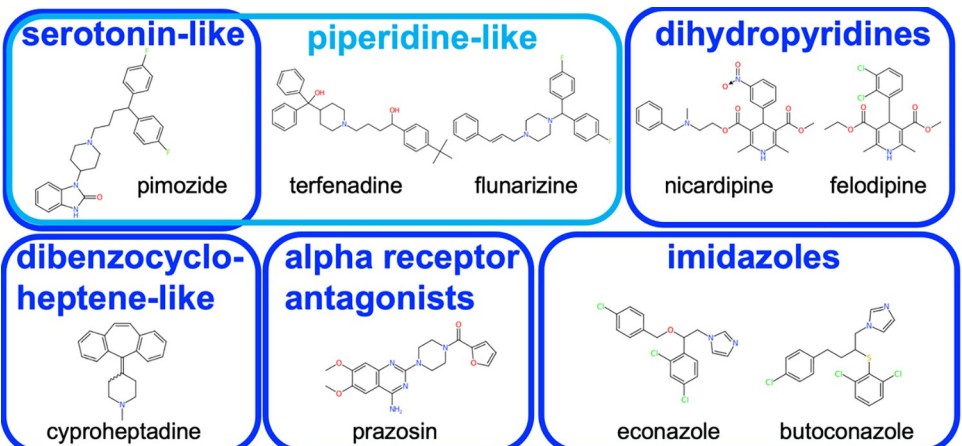

**Fig 3. The nine compounds tested *in vivo* in mice.** The images of compounds were generated using the CDKDepict website [45]. The salt form tested is given in **S2 Table**.

**Table 2. Worm burden reductions of *T. muris*-infected mice treated with drugs.**

| Treatment | Salt form tested in mice | Dosage tested in mice (mg/kg body weight) | No. of mice | Median worm burden | Worm burden reduction | P-value for worm burden reduction |
|---|---|---|---|---|---|---|
| control a (vehicle-only) | NA | NA | 6 | 12.0 | NA | NA |
| control b (vehicle-only) | NA | NA | 6 | 13.0 | NA | NA |
| control c (vehicle-only) | NA | NA | 6 | 21.0 | NA | NA |
| control d (vehicle-only) | NA | NA | 18 | 18.0 | NA | NA |
| control e (vehicle-only) | NA | NA | 18 | 20.0 | NA | NA |
| control f (vehicle-only) | NA | NA | 18 | 17.5 | NA | NA |
| mebendazole [a] | mebendazole | 50 mg/kg | 5 | 0.0 | 100.0% | 0.008 |
| mebendazole [b] | mebendazole | 50 mg/kg | 5 | 0.0 | 100.0% | 0.008 |
| mebendazole [c] | mebendazole | 50 mg/kg | 6 | 0.0 | 100.0% | 0.001 |
| pimozide [a] | pimozide | 35 mg/kg | 6 | 9.0 | 25.0% | 0.5 |
| pimozide [c] | pimozide | 35 mg/kg | 6 | 6.0 | 19.0% | 0.08 |
| pimozide [d] | pimozide | 35 mg/kg | 12 | 18.0 | - | 0.8 |
| pimozide [f] | pimozide | 70 mg/kg | 12 | 18.5 | - | - |
| nicardipine [a] | nicardipine hydrochloride | 50 mg/kg | 6 | 9.5 | 20.80% | 0.4 |
| nicardipine [e] | nicardipine hydrochloride | 50 mg/kg | 12 | 20.5 | - | - |
| terfenadine [a] | terfenadine | 100 mg/kg | 6 | 11.0 | 8.30% | 0.5 |
| felodipine [b] | felodipine | 35 mg/kg | 6 | 20.0 | - | - |
| prazosin [b] | prazosin hydrochloride | 100 mg/kg | 6 | 16.0 | - | - |
| econazole [b] | econazole nitrate | 50 mg/kg | 6 | 14.5 | - | - |
| flunarizine [c] | flunarizine dihydrochloride | 35 mg/kg | 6 | 21.5 | - | - |
| butoconazole [c] | butoconazole nitrate | 100 mg/kg | 6 | 18.5 | 12.0% | 0.2 |

The a, b, c, d, e in square brackets beside the drug names refer to the respective control groups. A one-sided Wilcoxon test was used to calculate the P-value for the worm burden reduction, comparing to the worm burden in the corresponding control worms. Note that one pimozide-treated mouse had to be culled early when treated with pimozide at 70 mg/kg.

### Except for mebendazole, drugs were not effective against T. muris *in mice*

In three independent experiments, the positive control mebendazole caused a 100% reduction in worm burden (one-sided Wilcoxon tests: *P* = 0.008, 0.008 and 0.001, respectively; **Table 2** and **Fig 4**). Pimozide reduced the worm burden by 19.0% in one experiment (see 'pim-c' versus 'ctl-c' in **Fig 4**), which was borderline statistically significant (one-sided Wilcoxon test: *P* = 0.08). However, two replicate experiments using the same dose of pimozide (35 mg/kg) did not show statistically significant reductions, nor did using a higher dose of pimozide (70 mg/kg). All mice dosed with cyproheptadine had to be culled due to adverse effects, and none of the other drugs showed statistically significant reductions in worm burden (**Table 2** and **Fig 4**).

## Discussion

Despite the millions of people worldwide affected by trichuriasis [1], and the pressing need for more effective drugs to treat whipworm infections, this disease receives little research

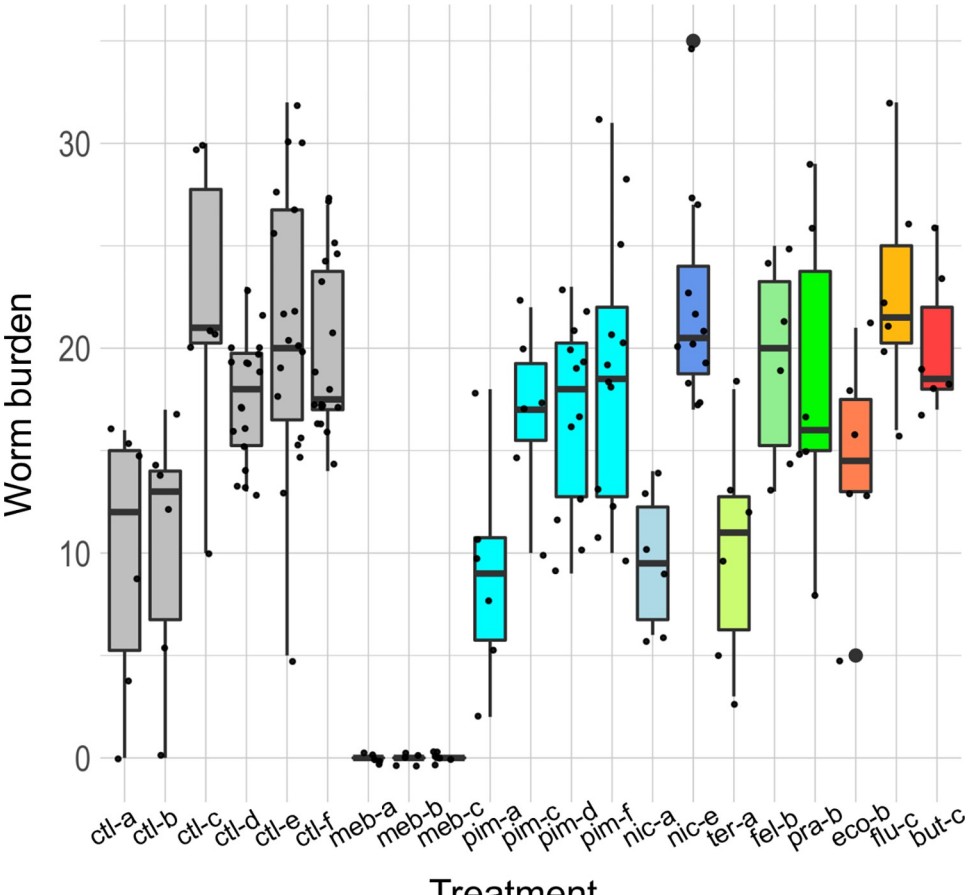

**Fig 4. Results of the *in vivo* screen in mice.** Each data point is one drug-treated or control mouse. The order of the treatments is the same as that in **Table 2**, and 'ctl' is vehicle-only control, 'meb' mebendazole, 'pim' pimozide, 'nic' nicardipine, 'ter' terfenadine, 'fel' felodipine, 'pra' prazosin, 'eco' econazole, 'flu' flunarizine, 'but' butoconazole. The letters 'a', 'b', 'c', 'd', 'e' beside the treatment names refer to the respective control groups.

attention, which makes it a major 'neglected tropical disease' [46]. Given the lack of investment in developing new drugs to treat trichuriasis, repurposing drugs that are currently approved for other uses is particularly attractive. While some drug repurposing libraries such as ReFRAME 'cast a wide net' by including all approved drugs as well as compounds that have undergone significant pre-clinical studies [10], we have used a comparative genomics approach to narrow our focus to a smaller set of drugs most likely to have key targets in whipworms. Specifically, we have prioritised whipworm targets that have *C. elegans/Drosophila melanogaster* orthologues with lethal/sterile phenotypes, and made use of curated data on drugs and their targets from ChEMBL [12] to predict 409 drugs that will target those whipworm proteins [11]. Our automated high-throughput screening platform INVAPP [13,14] enabled us to screen these 409 drugs at 100 μM against *T. muris* adults *ex vivo*. Our hit rate of ~12% (50/409) was relatively high compared to a previous rate of ~4% (20/480) for a library of drug-like small molecules screened against *T. muris* using INVAPP [18]. Thus, this demonstrated the utility of our comparative genomics approach to predict parasite proteins for targeting by approved drugs. Similarly, a screen of human oncology drugs against *T. muris* by Cowan *et al* 2016 [47] also found a relatively high hit rate (10.5%), presumably because *T. muris* has homologues of many of the human targets.

*T. muris* is relatively expensive and labour-intensive to maintain, so it is important to explore whether the same hits can be found by screening against *C. elegans*, which is far cheaper and easier to maintain. *C. elegans* is often used as a proxy for whipworms and other parasitic nematodes in drug screens [30,42], and functional assays in *C. elegans* have aided the development of new anthelmintics such as the amino-acetonitrile derivative monepantel [48]. However, for some screening libraries, there has been relatively low overlap reported between hits in *C. elegans* and a particular parasitic nematode (e.g. for the human hookworm *Ancylostoma ceylanicum* in Elfawal *et al* 2019 [29]). Indeed, when our drug library was screened against *C. elegans*, only two of the top 50 hits against whipworm were also hits against *C. elegans*. In fact, *C. elegans* had only 14 hits compared to 50 in *T. muris*. This may reflect a greater ability of *C. elegans* to detoxify the particular compounds in our screening library [49,50]. A limitation of our comparison between the *T. muris* and *C. elegans* screens is the longer compound exposure time used (*C. elegans* grown for six days from the L1 stage in the presence of compound at 20˚C before measurement). This may have led to false negatives in the *C. elegans* compound screen due to metabolism of an active compound or incomplete compound effect being followed by recovered *C. elegans* growth. Future work should investigate how the design of *C. elegans* screens could be optimised to model more closely the drug exposure time and worm biological activity of an *in vivo* parasitic nematode treatment assay and thus hopefully improve the predictive power of *C. elegans* assays.

Despite the low number of the top 50 *T. muris* hits also shared by *C. elegans* in our screen, 14 out of 50 have previously been reported as having activity against *C. elegans* and/or other nematodes in *ex vivo* screens that detected a variety of phenotypes (**Table 1**). For example, 7 of the 50 hits were previously reported in a screen against *Ancylostoma ceylanicum* L3s at 200 µM by Keiser *et al* 2016 [23], while 2 of the 50 hits were previously reported in a screen against *A. ceylanicum* adults at 30 µM by Elfawal *et al* 2019 [29] (**Table 1** and **S4 Fig**). This was the case for four of the nine highly active compounds that we tested in mice (pimozide, econazole, cyproheptadine, felodipine), suggesting these drugs may cause phenotypic effects in a broad range of nematodes.

For a promising *ex vivo* hit to be chosen as a lead compound for further drug development, one would require a worm burden reduction of at least ~60% *in vivo* [27]. Despite their high *ex vivo* activities, the nine drugs tested resulted in low and non-statistically significant worm burden reductions in mice. A previous screen against *T. muris* [47] also reported promising *ex vivo* hits, but low worm burden reductions (≤24%) for seven drugs tested in mice. Why do many drugs have high efficacy against *T. muris ex vivo*, but poor activities in mice? One possibility is that the ADME (absorption, distribution, metabolism, excretion) properties of the drugs mean that only a low concentration of drug (or a high concentration for only a short time) is available to *T. muris* in the large intestine. Cowan *et al* 2016 [47] pointed out that this has a high probability of occurring in repurposing screens, because most drugs approved for human use have been optimised for high absorption by the human gastrointestinal tract. For example, more than 50% of orally delivered pimozide is absorbed by the human gastrointestinal tract, compared to only 5–10% of mebendazole (from DrugBank [19]). Another possibility is that, even if a drug is available at a high concentration in the large intestine for a relatively long time, there may be poor uptake by *T. muris*. It is thought likely that the crucial pathway of mebendazole uptake into *T. muris* is by diffusion across the cuticle of the worms' posterior end, which lays freely in the intestinal tract [51], rather than uptake by the worm's long anterior end (including its mouth), which is embedded inside host intestinal epithelial cells. However, little is known about optimising this process for mebendazole or other drugs.

A bottleneck in drug discovery is *in vivo* testing in mice, which is expensive and sacrifices research animals. There is a clear need for development of intestinal *ex vivo* models and assays

to assess, further optimise and select hits, before performing *in vivo* testing. Based on the intestinal location of the whipworm parasite, and likely uptake of compounds by diffusion across the worms' posterior end, future screens for anti-whipworm compounds should try to increase understanding of how to optimise exposure of the worms to the drug, as this would likely improve the rate of success in mouse experiments. We suggest that one strategy would be to take the initial hits from a screen (of approved drugs and/or of other compounds) as leads, and then identify structural analogues that are predicted to have lower absorption (compared to the lead compounds) by the human gastrointestinal tract, by modifying their lipophilicity, molecular weight and polarity [52,53] (**S5 Fig**). While doing so, it would be important to avoid adding chemical fragments predictive of low bioaccumulation in nematodes (identified by Burns *et al* 2010 [50] for *C. elegans*), as our goal is to identify compounds that like mebendazole are taken up well by the worm but not by the human gastrointestinal tract. The anti-whipworm activity of the structural analogues could be assessed in a second *ex vivo* screen, and their absorption by the human gastrointestinal tract predicted using human intestinal organoids or cell lines [8,52]. After optimising for low absorption, top compounds could then be assessed for anti-whipworm activity against *T. muris* embedded in intestinal organoids, to determine whether compounds are readily taken up by and have a phenotypic effect on *T. muris* embedded in the host gastrointestinal tract [8]. Strong hits from this latter step could be tested in mice. By adding these additional steps in the drug discovery pipeline for trichuriasis, we will better understand how to increase exposure of the worms to the compounds, and this should help us to better select compounds likely to give a higher worm burden reduction in mice. This analogue-based strategy loses the 'fast-track to the clinic' benefit of direct repurposing, but may be necessary for trichiuriasis because most approved drugs have been optimised for human bioavailability and are therefore rapidly absorbed by the host gastrointestinal tract. However, a bonus of optimising compounds for low absorption by the human gastrointestinal tract would be reduced side-effects and tolerance of higher doses by patients.

## Supporting information

**S1 Table. The 409 compounds obtained for the initial screen.**
(XLSX)

**S2 Table. The 9 approved drugs that were tested in mice.**
(XLSX)

**S3 Table. Results for the top 50 hits from the screen at 100 μM for 24 hours.**
(XLSX)

**S4 Table. The 71 additional compounds obtained for the screen, which were structurally related to our top 50 hits in the initial screen.**
(XLSX)

**S5 Table. The 14 approved drugs that have estimated EC$_{50}$ values of ≤50 μM.**
(XLSX)

**S1 Movie. Recordings of *T. muris* adults treated with DMSO alone, or with astemizole or pimozide at 100 μM for 24 hours.** https://doi.org/10.6084/m9.figshare.22193977.v1
(AVI)

**S1 Fig. The top 50 hits in our initial *ex vivo* screen.** The images of compounds were generated using the CDKDepict website [45]. The salt form tested is given in **S3 Table**.
(TIF)

**S2 Fig. Searching for compounds with substructures present in our top 50 hits.** The substructures that were used in the 'substructure search function' in DataWarrior, to search for additional approved drugs with substructures present in our top 50 hits. Images of compounds were generated using the CDKDepict website [45].
(TIF)

**S3 Fig. The top 17 hits in our screen of additional approved drugs that were structurally related to our initial top 50 hits.** The images of compounds were generated using the CDKDepict website [45]. The salt form tested is given in **S4 Table**.
(TIF)

**S4 Fig. Overlaps between hits in different drug screens.** (a) Overlap between our hits in *C. elegans* and in *T. muris*, after re-screening at 100 μM, (b) overlap between our hits in *T. muris*, and those of Keiser *et al* 2016 [23] in *Ancylostoma ceylanicum*, (c) overlap between our hits in *T. muris*, and those of Elfawal *et al* 2019 [29] in *A. ceylanicum*.
(TIF)

**S5 Fig. A suggested pipeline for drug discovery for trichuriasis.**
(TIF)

## Acknowledgments

We thank Fiona Hunter, Prudence Mutowo and Andrew Leach from ChEMBL for helpful advice on ChEMBL data; Noel O'Boyle for cheminformatics advice; and Kathryn Else and Jennifer Keiser for advice on *in vivo* testing. We are grateful to Olga Woolmer and Selina Hopkins for helpful discussions and advice on mouse welfare and regulatory compliance. We thank other members of the Berriman and Sattelle teams for useful discussions, especially Mandy Sanders. For the purposes of Open Access, the author has applied a CC BY public copyright licence to any Author Accepted Manuscript version arising from this submission.

## Author Contributions

**Conceptualization:** Avril Coghlan, Frederick A. Partridge, María Adelaida Duque-Correa, Gabriel Rinaldi, David B. Sattelle, Matthew Berriman.

**Data curation:** Avril Coghlan, Frederick A. Partridge.

**Formal analysis:** Avril Coghlan, Frederick A. Partridge, Sirapat Tonitiwong.

**Funding acquisition:** David B. Sattelle, Matthew Berriman.

**Investigation:** Avril Coghlan, Frederick A. Partridge, María Adelaida Duque-Correa, Gabriel Rinaldi, Simon Clare, Lisa Seymour, Cordelia Brandt, Tapoka T. Mkandawire, Catherine McCarthy, Marina Nick, Anwen E. Brown, Sirapat Tonitiwong.

**Methodology:** Avril Coghlan, Frederick A. Partridge, María Adelaida Duque-Correa, Gabriel Rinaldi, Simon Clare, Lisa Seymour, Cordelia Brandt.

**Project administration:** Avril Coghlan, Frederick A. Partridge, María Adelaida Duque-Correa, Gabriel Rinaldi, Nancy Holroyd.

**Resources:** David B. Sattelle, Matthew Berriman.

**Software:** Avril Coghlan, Frederick A. Partridge.

**Supervision:** Frederick A. Partridge, María Adelaida Duque-Correa, Gabriel Rinaldi, Simon Clare, David B. Sattelle, Matthew Berriman.

**Validation:** Frederick A. Partridge.

**Visualization:** Avril Coghlan, Frederick A. Partridge.

**Writing – original draft:** Avril Coghlan.

**Writing – review & editing:** Avril Coghlan, Frederick A. Partridge, María Adelaida Duque-Correa, Gabriel Rinaldi, Nancy Holroyd, David B. Sattelle, Matthew Berriman.

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
