## [Decision Letter · Decision Letter 0]

10 Apr 2023

Dear Dr. Coghlan,

Thank you very much for submitting your manuscript "A drug repurposing screen for whipworms informed by comparative genomics" for consideration at PLOS Neglected Tropical Diseases. As with all papers reviewed by the journal, your manuscript was reviewed by members of the editorial board and by several independent reviewers. The reviewers appreciated the attention to an important topic. Based on the reviews, we are likely to accept this manuscript for publication, providing that you modify the manuscript according to the review recommendations. 

Sincerely,

Mostafa Zamanian, Ph.D.

Guest Editor

Eva Clark

Section Editor

Reviewer's Responses to Questions

**Key Review Criteria Required for Acceptance?**

**Methods**

-Are the objectives of the study clearly articulated with a clear testable hypothesis stated?

-Is the study design appropriate to address the stated objectives?

-Is the population clearly described and appropriate for the hypothesis being tested?

-Is the sample size sufficient to ensure adequate power to address the hypothesis being tested?

-Were correct statistical analysis used to support conclusions?

-Are there concerns about ethical or regulatory requirements being met?

Reviewer #1: The study design is appropriate and the correct statistical analysis is performed. There are no concerns regarding regulatory requirements.

Reviewer #2: see below

Reviewer #3: See the review report the summary and General Comments

**Results**

-Does the analysis presented match the analysis plan?

-Are the results clearly and completely presented?

-Are the figures (Tables, Images) of sufficient quality for clarity?

Reviewer #1: Unless I am missing it, there doesn’t appear to be a figure on the C. elegans ex vivo drug screen? These data are interesting and the paper would benefit from adding a figure summarizing these results. This result seems consistent with a previously report for ex vivo hookworm screen (PMID: 31451730) where C. elegans had limited overlap.

Reviewer #2: see below

Reviewer #3: See the review report in summary and General Comments

**Conclusions**

-Are the conclusions supported by the data presented?

-Are the limitations of analysis clearly described?

-Do the authors discuss how these data can be helpful to advance our understanding of the topic under study?

-Is public health relevance addressed?

Reviewer #1: The conclusions are supported by the data which is clearly presented.

Reviewer #2: see below

Reviewer #3: See the review report in Summary and General Comments.

**Editorial and Data Presentation Modifications?**

Reviewer #1: Minor points:

1. Is motility an antiparasitic mechanism for drugs acting on whipworms, or is the goal to actually kill the worm? If paralysis is not necessarily a sufficient outcome (since the anterior of the worm is embedded in the intestinal mucosa and so the worm may not wash away), it may be useful to reassay motility after drug washout. I bring this up not as a request for more data, but a possible discussion point. But if the authors are aware of whether their ex vivo hits such as aminergic compounds were actually killing or just paralyzing the worms, this would be interesting to comment on.

2. The authors note that in compound selection they made a choice to discard annotated antipsychotics. I wonder if this is a decision that should be made during compound selection because several hits seem to either be antipsychotics or structurally very similar to antipsychotics. Terfenadine is similar in structure to antipsychotic haloperidol and numerous tricyclic antidepressants are still in the list (ex. chlorprothixene, and cyproheptadine is closely related to TCAs). Pimozide which was a hit and tested in mice and is an antipsychotic. Is this still a triage step that should be used going forward?

3. Perhaps in the legend of somewhere the putative nematode targets of ex vivo hit compounds could be elaborated on? This may be in the 2019 paper, but I was wondering how antifungals (econazole, butoconazole) or surfactants / detergents (nonoxynol-9, thonzonium) may be acting on worms.

4. The authors refer to high hit rate in primary screen (12%) supporting utility of a comparative genomics approach to compound selection. A question in my mind when reading was to what degree this improved over a baseline for a typical screen. Perhaps worth expanding upon by citing references of other whipworm screens (understanding that screening methodology may of course vary)?

5. Would optimizing for low absorption across the host GI tract also impair uptake across nematode tissues? Would more polar, less lipophilic compounds decrease diffusion into nematode?

Reviewer #2: see below

Reviewer #3: See the review report in Summary and General Comments

**Summary and General Comments**

Reviewer #1: There is a need for new anthelmintics to treat whipworm infections, given that drugs such as mebendazole can be subcurative. This manuscript reports the results of a medium throughput drug screen on T. muris, with the aim of identifying new therapies or lead anthelmintic compounds. Specifically, authors chose these based on comparative genomics approach focusing on small molecules likely to act on key whipworm targets. The data are clearly presented and the rational for the workflow of the screen progressing from ex vivo primary screen to testing potency to dosing mice in vivo is clearly articulated. While none of the ex vivo hits displayed in vivo efficacy, these data are still quite useful to the community and will inform the design of future screening efforts. The authors propose reasonable explanation for the discrepancy between the ex vivo and in vivo results, noting that many compounds optimized for high host bioavailability (uptake from GI) when in fact whipworm may require opposite if diffusion across posterior of animal is main route of parasite exposure. The main area I would like to see some more elaboration on in the results is the C. elegans screening data.

Reviewer #2: The Coghlan et al manuscript is a very quick, easily absorbed manuscript that I found to be insightful, especially in the discussion’s commentary. I do think that PNTD is an appropriate venue for its publication. I have only minor suggestions for improvement.

1. I am not sure the last sentence in the abstract is appropriate for that section (too prescriptive for an abstract?) but I leave that to the authors to consider further.

2. Table 1, column G has some nice citations showing previous work on the indicated compounds against helminths. However, I would also be interested to see the overlap (and non-overlap) of molecules screened and hits obtained in the Venn diagram with the authors current work and that of Keiser et al 2016 and Elfawal et al., 2019 (PMID 31451730).

3. It would be convenient to interested readers to have a supplementary table (or modify Sup Table 1) that has all examined molecules (409+71) and their in vitro results, whether or not they are considered ‘hits’, and their in vivo results, all in different columns. This helps for others to analyzed the data for comparison purposes, although all of the releveant data is present, but in different files and slightly different formats etc.

4. It might be useful to the read to have a few more sentences in the introduction that describes in more detail how the 817 compounds were identified, and how the 409 molecules were selected as a subset.

5. Another good reference for the ADME properties in elegans to go along with Lindblom and Dodd 2006 in the discussion is the Burns et al 2010 Nat Chem Bio paper (PMID 20512140).

6. I found the last paragraph of the discussion of particular interest- perhaps it deserves its own pipeline figure? While I do like the ideas, and they deserve articulation, I do think the authors should consider additional issues that go along with the suggested pipeline. The great thing about screening for established drugs for off-labeled use is that everyone, in principle, is behind that. The drug companies potentially make more money with expanded use, and there can be fast-track to the clinic. However, when folks start analoging, there may be patent issues to be dealt with- their will likely be infringement against the patents of the original drug (because they are typically broad), or if the patents have run out, then the analog is in the public domain and industry has no incentive to develop the compound further. A second issue is that there may be no fast track to the clinic with an analog of a well characterized and approved drug.

If the authors can envision routes around these issues, they should articulate those routes. If the authors can’t envision ways around these issues, they should discuss the limitations of bringing an effective molecule to market. Regardless, I find it all very interesting either way.

7. Minor note- Figure 1, fourth blue box, I think ‘too’ can simply be removed.

8. At some point in the future (but beyond the scope of this manuscript), it might of interest to the authors to test the nematode effective analog, nemadipine, in vitro and in vivo given the the overall effectiveness of the DHPs.

Reviewer #3: Review of Avril Coghlan et al

“A drug repurposing screen for whipworms informed by comparative genomics”

The manuscript is interesting, helpful, informative, well-written, and easy to read. In a previous report, using comparative genomic tools, the authors identified 409 drugs approved for human use predicted to target essential parasitic worm proteins based on homology to known druggable targets in other species. In this report, the authors tested the possibility of repurposing these 409 compounds towards parasitic nematodes using initial ex vivo screening, dose-response, and in vivo testing. The paper addresses a critical issue in drug discovery against parasitic nematodes: the disagreement between ex vivo potency and in vivo efficacy. Even though the investigators identified 14 compounds with ex vivo potency, none showed significant in vivo activity when tested in mice infected with the murine whipworm parasites. The authors give a detailed discussion on the reason behind such discrepancy and offer suggestions to modify the whipworm screening pipeline to better predict in vivo activities in mice. 

The paper is suitable for publication after significant revision addressing and commenting on the following issues. 

Introduction: 

The introduction is informative but relatively light. It may benefit the general readers if the authors address the challenges facing drug discovery in parasitic nematodes, including, but not limited to, the lack of culture systems in parasites and the expectation/requirement that final candidates must show reasonable activity against multiple species of human gastrointestinal nematodes. It is worth noting why the authors, among all STH, focused their screening against whipworm parasites, as new drugs are equally urgently needed to replace benzimidazoles due to the spreading resistance. 

Methods:

The methods section needs careful revision as many of the listed assays need more essential details. The initial screen at 100µM is relatively high, and the authors need to comment on how they determined the screening dose. 

C. elegans is a free-living nematode, and it is unsuitable to call the assay Ex vivo, as the term is more specific to tissues or parasites tested outside the host body. So far, thousands of strains have been generated with multiple wildtypes isolates already available in C. elegans. The authors need to give more information on the strain of C. elegans they used. How many L1 were tested per replicate? 6 days of L1 assay is quite long when incubated at 20c for two reasons. First, at this temperature, for example, in the N2 strain take about three days to develop into adults-again, depending on the strain used. If a single L1 survives, the treatment will eventually grow into an adult and reproduce many F1s. which could lead to a higher false negative rate, especially if the readout is cumulative motility. Second, six days assay could result in an increased rate of drug metabolism. For example, suppose a specific drug induces paralysis or a reversible kind of activity. In that case, the treated worms will gain viability after metabolizing the drug, leading to a higher rate of false negatives, and scoring the assay plates when the control wells at the early L4 stage, about 44 hours, could be better here. 

Assays with multiple replicates are very useful when motility is the only readout, as the range of motility in whipworms is quite large, even in the control group. For example, male whipworms are much smaller and generate lower pixel change than females. When testing the 71 additional approved drugs for prioritization, the authors tested four in total worms, one worm at 100µM, two worms at 50µM, and one worm at 20µM. Single or double-worm motility readout will not generate useful or reliable information. The lack of reasonable replication gives the impression that the 17 compounds selected for retesting at 100µM were chosen arbitrarily. 

In mice's in vivo drug screening, the authors inoculated each mouse with 20 embryonated eggs. After 35 days, they collected fecal material and generated fecal smears to check for the presence of T. muris eggs. It needs to be clarified here how the smears were prepared. The standard practice in the field is to use a fecal egg counting method to count the number of eggs per gram of feces. Also, more than qualitative checking of the presence of T. muris eggs in infected mice is needed for well-designed in vivo studies. A better practice is to count the number of eggs per gram of feces, then group animals into experimental groups with similar averages of the pre-treatment fecal egg count. 

Mice were treated for three days starting on day 36 p.i., and animals were euthanized on day 44 p.i. to count the worm burdens. The authors didn’t report any post-treatment fecal egg count, a very important readout in the parasitic nematode in vivo studies. Especially in early investigation phases, candidate drugs may fail to induce worm expulsion but are effective enough to reduce egg-laying capacity significantly, an activity that could be improved.

 Results

The result section is easy to read and straightforward, and the figures and tables are well-designed and easy to follow. However, some issues need to be addressed here. 

This header, “Pimozide was the most efficacious drug tested against T. muris in mice,” is inaccurate and may be misleading, as a 19% reduction in motility is not nearly considered the most efficacious nor considered active at all. 

In Figure 2 (E), the first figure, “Astemizole EC50 value needs to be corrected as it now reads 14 ± 48µM. 

In Figure 4, the data show that many individual mice had a worm burden above 20 worms, even though these mice were inoculated with only 20 embryonated eggs. The authors must comment on discrepancies between the results and methods reporting the in vivo studies in mice sections.

Discussions

the discussion is interesting, convincing, and makes sense. In the section discussing the overlap between C. elegans and whipworms, the authors referred to the suitability of C. elegans as a surrogate model for whipworms. However, their data doesn’t support that claim, as only two of the 50 compounds active against whipworms overlap with C. elegans. A similar finding was reported recently when testing the quality of C. elegans in predicting antiparasitic nematode activity and found that C. elegans is prone to a higher rate of false negatives. I suggest that the authors reference opinions that argue against the suitability of C. elegans in modeling parasitic nematodes. 

The authors greatly discussed the reasons behind the discrepancy between ex vivo potency and in vivo activity. They offered exciting tips for dealing with this critical problem challenging the anthelmintic drug discovery efforts.

PLOS authors have the option to publish the peer review history of their article (what does this mean?). If published, this will include your full peer review and any attached files.

Reviewer #1: Yes: John Chan

Reviewer #2: No

Reviewer #3: No

Figure Files:

Data Requirements:

Reproducibility:

References

---

## [Editor Report · Decision Letter 1]

6 Jul 2023

Dear Dr. Coghlan,

We are pleased to inform you that your manuscript 'A drug repurposing screen for whipworms informed by comparative genomics' has been provisionally accepted for publication in PLOS Neglected Tropical Diseases.

Best regards,

Mostafa Zamanian

Guest Editor

Eva Clark

Section Editor

---

## [Editor Report · Acceptance letter]

24 Aug 2023

Dear Dr. Coghlan,

We are delighted to inform you that your manuscript, "A drug repurposing screen for whipworms informed by comparative genomics," has been formally accepted for publication in PLOS Neglected Tropical Diseases.

Best regards,

Shaden Kamhawi

co-Editor-in-Chief

Paul Brindley

co-Editor-in-Chief
